# Rosmarinic Acid and Sodium Citrate Have a Synergistic Bacteriostatic Effect against *Vibrio* Species by Inhibiting Iron Uptake

**DOI:** 10.3390/ijms222313010

**Published:** 2021-12-01

**Authors:** Peng Lu, Miaomiao Sui, Mimin Zhang, Mengyao Wang, Takehiro Kamiya, Ken Okamoto, Hideaki Itoh, Suguru Okuda, Michio Suzuki, Tomiko Asakura, Toru Fujiwara, Koji Nagata

**Affiliations:** 1Department of Applied Biological Chemistry, Graduate School of Agricultural and Life Sciences, The University of Tokyo, 1-1-1 Yayoi, Bunkyo-ku, Tokyo 113-8657, Japan; suimiaomiao1989@outlook.com (M.S.); zhangmimin001@hotmail.com (M.Z.); wmy-puppet@outlook.com (M.W.); akamiyat@g.ecc.u-tokyo.ac.jp (T.K.); akenokamoto@g.ecc.u-tokyo.ac.jp (K.O.); itohh@g.ecc.u-tokyo.ac.jp (H.I.); okuda@g.ecc.u-tokyo.ac.jp (S.O.); amichio@g.ecc.u-tokyo.ac.jp (M.S.); asakura@g.ecc.u-tokyo.ac.jp (T.A.); atorufu@g.ecc.u-tokyo.ac.jp (T.F.); 2Agricultural Bioinformatics Research Unit, Graduate School of Agricultural and Life Sciences, The University of Tokyo, 1-1-1 Yayoi, Bunkyo-ku, Tokyo 113-8657, Japan

**Keywords:** ferric ion-binding protein, vibriosis, *Vibrio* species, spice extracts, rosmarinic acid, bacteriostatic agent

## Abstract

Background: New strategies are needed to combat multidrug-resistant bacteria. The restriction of iron uptake by bacteria is a promising way to inhibit their growth. We aimed to suppress the growth of *Vibrio* bacterial species by inhibiting their ferric ion-binding protein (FbpA) using food components. Methods: Twenty spices were selected for the screening of FbpA inhibitors. The candidate was applied to antibacterial tests, and the mechanism was further studied. Results: An active compound, rosmarinic acid (RA), was screened out. RA binds competitively and more tightly than Fe^3+^ to VmFbpA, the FbpA from *V. metschnikovii*, with apparent *K*_D_ values of 8 μM vs. 17 μM. Moreover, RA can inhibit the growth of *V. metschnikovii* to one-third of the control at 1000 μM. Interestingly, sodium citrate (SC) enhances the growth inhibition effect of RA, although SC only does not inhibit the growth. The combination of RA/SC completely inhibits the growth of not only *V. metschnikovii* at 100/100 μM but also the vibriosis-causative pathogens *V. vulnificus* and *V. parahaemolyticus*, at 100/100 and 1000/100 μM, respectively. However, RA/SC does not affect the growth of *Escherichia coli*. Conclusions: RA/SC is a potential bacteriostatic agent against *Vibrio* species while causing little damage to indigenous gastrointestinal bacteria.

## 1. Introduction

*Vibrio* is a genus of ubiquitous bacteria that inhabit a wide range of aquatic and marine environments and cause seafood contamination and infectious diseases in humans. More than 30 species of *Vibrio* have been discovered to date and *V*. *cholerae*, *V*. *parahaemolyticus*, *V*. *alginolyticus*, *V*. *vulnificus*, *V*. *fluvialis*, *V*. *hollisaea*, *V*. *mimicus*, and *V*. *metschnikovii* are among the most frequently isolated human pathogens. These bacteria are found in environments of moderate or high salinity, including in seawater and seafood, and they are agents of human vibriosis [1].

Sushi is a Japanese cuisine that is popular worldwide. People consume a large number of raw marine fishes, crustaceans, and shellfishes as sushi with soy sauce. However, eating raw seafood contaminated by microorganisms may cause disease; indeed, *Vibrio*-related seafood-borne diseases are rapidly increasing [2]. According to the Intergovernmental Panel on Climate Change (IPCC), on a global scale, the ocean surface is warming due to climate change. The upper 75 m near the ocean surface warmed by 0.11 °C (0.09 °C to 0.13 °C) per decade from 1971 to 2010 [3]. Since 2012, The European Food Safety Authority (EFSA) has been highlighting the emerging threat of *Vibrio* resulting from climate change. The increased surface seawater temperature and increased nutrient input in the ocean have spread *Vibrio*, causing outbreaks of seafood contamination [4].

According to the U.S. Centers for Disease Control and Prevention (CDC), the peak of *Vibrio* infection occurs from May to October, during which the climate is warmer than the rest of the year [5]. Such a warm season is generally recognized as the “*Vibrio* season” because it is suitable for *Vibrio* growth. Furthermore, this season is expanding into the all months, and the risk of infection is increasing [5]. From 1996 to 2010, a total of more than 7000 cases of *Vibrio* infection were reported in the United States [6]. However, because *Vibrio* bacteria are not easily identified with routine testing, there are even more cases not reported.

At present, CDC estimates that approximately 80,000 people per year are infected with *Vibrio* in the United States, of which about 52,000 of the infections are estimated to be the result of eating contaminated seafood. Every year, about 100 people die from the infection [7]. Therefore, it is urgent to take effective actions to decrease the risk of infection by *Vibrio* globally.

At the initial stage of this study, we targeted an infectious Gram-negative bacterium: *V*. *metschnikovii*. Although *V*. *metschnikovii* is less frequently reported than *V*. *cholera*, *V*. *parahaemolyticus*, and *V*. *vulnificus*, it will also be a potential emerging issue [8]. 

The abuse of non-biodegradable antibiotics in the food industry has promoted their persistence in aquatic environments for years. The direct cause for this is the development of antibiotic-resistance bacteria pathogenic to humans, fish, and other aquatic animals [9]. The emergence of antibiotic-resistance is threatening the usefulness of new bactericidal agents and suppressing drug development [10]. Therefore, there is a need to develop a new antimicrobial agent beyond antibiotics for the inhibition of *V*. *metschnikovii* growth.

Iron is a vital element in almost all living organisms. In the case of *Vibrio*, iron is the growth-limiting factor in nutrient-rich ocean environments. Low iron levels not only reduce the rate of growth of *V*. *metschnikovii* but also affect its survival and ability to persist [11]. As an alternative to antibiotics, the restriction of iron uptake is a potential antibacterial therapy [12]. To support general biological functions in bacteria, around 10^−6^ to 10^−7^ molar iron is required per cell [13]. Ferric ion-binding protein (Fbp) A is a periplasmic-binding protein (PBP) from an ATP-binding cassette (ABC) transporter in Gram-negative bacteria, particularly for binding un-chelated or naked Fe^3+^ solubilized in the solution. This FbpA efficiently takes up iron together with the inner membrane transporter, FbpBC. Since FbpA is highly conserved among Gram-negative bacteria, FbpAs from various species are chosen as antibacterial targets and have been structurally or functionally studied [14]. Moreover, the antisense peptide nucleic acid of the FbpA gene has been reported to have bactericidal activity toward *Neisseria meningitidis* [15].

As a Gram-negative bacterium, *V*. *metschnikovii* has an FbpA termed VmFbpA. VmFbpA is responsible for Fe^3+^ binding in the periplasm and delivering Fe^3+^ into the cytoplasm by docking at the transmembrane domain and has a sequence similarity of ≈30% to the FbpAs of other pathogenic *Vibrio* species. Consequently, VmFbpA is a promising target protein for iron limitation.

As part of the Japanese culture of eating raw seafood, complementary ingredients are typically used to enhance the flavor. Spices are commonly introduced as flavor enhancers in the food industry, which is a practice that is widely accepted by the general population. This study was original because we inhibited bacterial growth by limiting iron uptake, rather than by using conventional antibiotics. Therefore, to lower the risk of oral infections by *Vibrio* species, specific spice extracts or chemicals that can both enhance the flavor of raw seafood and inhibit FbpA activity, to control the growth of *Vibrio* species, should be identified.

Here, we report a natural inhibitor of VmFbpA from rosemary extract, rosmarinic acid (RA), which was screened out from 20 spices, and show that its antibacterial effect on several *Vibrio* species is enhanced in the presence of sodium citrate (SC).

## 2. Results

### 2.1. Screening of VmFbpA Inhibitors from Aqueous Extracts of 20 Spices

Assays for the inhibition of Fe^3+^ binding of VmFbpA were performed, and the remaining Fe concentrations were measured via ICP-MS using aqueous extracts of 20 spices at a final concentration of 2.5 mg/mL (Figure 1a). Only cinnamon and rosemary extracts decreased the remaining Fe concentration by more than 50% compared to the Fe^3+^-VmFbpA control group. Furthermore, no spice extracts increased the remaining Fe concentration higher than the Fe^3+^-VmFbpA control group. Therefore, cinnamon and rosemary extracts were considered candidate inhibitors of VmFbpA.

### 2.2. Spectral Analysis of VmFbpA Interacting with RA

To assess their dose-dependent effects on the inhibition of Fe^3+^ binding by VmFbpA, different concentrations (0, 0.9, 1.9, 3.8, 7.5, and 15 mg/mL) of cinnamon and rosemary extracts were subjected to inhibition assays using UV-visible absorbance spectroscopy (Figure 1b–e). In the case of cinnamon extracts, the color of the eluents changed from yellow to colorless and finally became dark red as the applied concentration increased (Figure 1b). In the case of rosemary extracts, the color of the eluents changed from yellow to colorless and finally became yellow again (this yellow color is the color from rosemary extracts but not Fe^3+^) as the applied concentration increased (Figure 1d). A significant peak shift (from ≈412 nm to ≈380 nm) was observed in the spectra of the samples treated with rosemary extracts (Figure 1e). However, no such peak shift was observed in the samples treated with cinnamon extracts (Figure 1c). Consequently, rosemary extracts were selected for further analysis.

Second, a major component of rosemary extract, RA (0, 0.15, 0.30, 0.60, 1.2, and 2.4 mg/mL), was subjected to an inhibition assay. Compared to the spectrum of VmFbpA or RA only, that of VmFbpA mixed with RA showed a specific peak (Figure 1f,g). In addition, a similar peak shift (from ≈412 nm to 380 nm) to that caused by the rosemary extract was observed (Figure 1e).

### 2.3. Quantitative Analysis of the Inhibition of Fe^3+^ Binding to VmFbpA by Rosemary Extracts and RA

Rosemary extracts at 0, 0.9, 1.9, 3.8, 7.5, and 15 mg/mL, and RA at 0, 6.5, 13.0, 26.0, 52.0, 104.0, 208.0, 416.3, 832.6, 1665.2, 3330.5, and 6661.0 μM, were used to determine the IC_50_ value (Figure 2a–d). The IC_50_ of rosemary extracts was 2.4 ± 0.3 mg/mL and the IC_50_ of RA was 800 ± 100 μM (0.29 ± 0.04 mg/mL).

The reduction of Fe^3+^ to Fe^2+^ by RA was evaluated by mixing 40 μM FeCl_3_ with various concentrations of RA (Figure 2e,f). A dose-dependent effect of RA in reducing Fe^3+^ to Fe^2+^ was observed. According to the regression curve, 1 μM RA can reduce ≈2.44 μM Fe^3+^ to Fe^2+^.

The affinity between RA/Fe^3+^ and VmFbpA was measured via ITC (Figure 3a–c). The best fit of the ITC data in Figure 3a was attained with a single-site model; the apparent association constant (*K*_A, app_) was 1.3 × 10^5^ ± 0.2 × 10^4^ M^−1^. The apparent *K*_D_ of RA binding to VmFbpA was ≈8 μM (Figure 3a), which is two-fold stronger than Fe^3+^ binding to VmFbpA (apparent *K*_D_ ≈ 17 μM) (Figure 3c). However, a more complicated curve was obtained for ITC experiments using Fe^3+^-VmFbpA, which could not be fit to any model (Figure 3b). The thermodynamic parameters for the reaction between RA and apo VmFbpA were Δ*H* = −15 ± 9 kcal/mol and Δ*S* = −0.04 ± 0.03 kcal/mol/deg, which are shown in Figure 3d.

### 2.4. Spectra Analysis of Fe^3+^ Interacting with RA

The interactions between RA and Fe^3+^ were also investigated (Appendix A), including the UV-visible absorbance spectra of RA supplemented with different concentrations of FeCl_3_ (Appendix A). After subtraction of the respective FeCl_3_ spectra (Appendix A), the net spectra were obtained (Appendix A). The net spectra showed two peaks (≈290 nm and ≈320 nm) in the spectrum of 20 μM RA, whereas only one peak at ≈250 nm was observed with supplementation of 20, 30, and 40 μM FeCl_3_. Moreover, the spectrum of 20 μM RA with supplementation of 10 μM FeCl_3_ showed an intermediate pattern during the peak shift from ≈290 nm and ≈320 nm to ≈250 nm.

### 2.5. Inhibition of Bacterial Growth by RA

The bacteriostatic effect of RA on the growth of *V*. *metschnikovii* was assayed (Figure 4). The growth of *V*. *metschnikovii* in nutrient broth (NB) medium leads to a final OD_600_ value of ≈0.6 after 5 h using a microplate spectrophotometer (Benchmark Plus, Bio-Rad). In the presence of 500 μM RA, the final OD_600_ value decreased to about one-half of the control group. In the presence of 1000 μM RA, the final OD_600_ value decreased to about one-quarter of the control group (Figure 4a). The OD_600_ value at 7 h was used for statistical analysis. In the presence of 100 μM RA, the OD_600_ value at 7 h showed a statistically significant decrease (*p* = 0.026) compared to the control group. The OD_600_ value at 7 h further decreased as the RA concentration increased to 500 μM and 1000 μM. In the presence of 500 μM RA, the OD_600_ value at 7 h decreased to ≈0.4 (*p* < 0.001) and in the presence of 1000 μM RA, the OD_600_ value at 7 h decreased to ≈0.18 (*p* < 0.001) (Figure 4b).

The growth of *E*. *coli* BL21(DE3) and KRX in NB medium leads to a final OD_600_ value of ≈0.8 after 5 h using a microplate spectrophotometer (Benchmark Plus, Bio-Rad, Hercules, CA, USA). However, the OD_600_ value and the bacterial growth curve were not affected by RA at 25 μM to 1000 μM (Figure 4c,d).

### 2.6. Iron Utilization in V. metschnikovii

The weight of *V*. *metschnikovii* after 7 h culture in NB medium decreased with increasing RA concentration (Figure 5a). The iron consumed by *V*. *metschnikovii* in all samples showed the same tendency, as measured using ICP-MS. The Fe utilization ratio decreased to half of the initial value after adding 100 μM RA (*p* = 0.05) and further decreased to a quarter as the RA concentration increased to 1000 μM (*p* < 0.01) (Figure 5b). The recovery of *V*. *metschnikovii* in the presence of 1000 μM RA was investigated by supplementation of 10 and 50 μmol FeCl_3_ to final concentrations of 1 μM and 5 μM. When *V*. *metschnikovii* was supplemented with 1 μM FeCl_3_, slight recovery was observed, whereas after supplementation with 5 μM FeCl_3_, the recovery of *V*. *metschnikovii* was more obvious (Figure 5c). Based on the OD_600_ value at 7 h, the groups supplemented with 1 μM (*p* < 0.01) and 5 μM FeCl_3_ (*p* < 0.001) were significantly different from the negative control (Figure 5d).

### 2.7. Inhibition by RA and Sodium Citrate (SC) of the Growth of V. metschnikovii

The growth of *V*. *metschnikovii* was inhibited to ≈70% by 1000 μM RA, and this inhibition was enhanced by 10, 100, and 1000 μM SC (Figure 6a). Interestingly, the growth of *V*. *metschnikovii* was hardly inhibited by 10, 100, and 1000 μM SC without adding 1000 μM RA. The recovery of *V*. *metschnikovii* growth was observed as the RA concentration decreased from 1000 to 25 μM RA in the presence of 100 μM SC (Figure 6b).

The addition of RA or using the combination of RA and SC did not have a significant inhibitory effect on *E*. *coli* growth; all of the *E*. *coli* groups had an OD_600_ value of ≈0.8 after incubation for 7 h in NB medium (Figure 6c,d).

### 2.8. Mechanism of RA Binding to VmFbpA

The crystal structure of apo VmFbpA was determined by X-ray crystallography (Appendix A). Diffraction-quality crystals of apo VmFbpA were obtained with a reservoir composition of 0.25 M ammonium tartrate dibasic, 25% PEG3350, 100 mM Tris-HCl (pH 7.0). The apo VmFbpA crystal belonged to space group *P*6_3_22, with unit-cell parameters *a* = 90.79, *b* = 90.79, and *c* = 149.68 (Appendix A). The crystal structure of apo VmFbpA was determined at 1.86 Å resolution by molecular replacement using the structure of TtFbpA (PDB entry 3WAE) as the search model. The final structure of apo VmFbpA contained one monomeric VmFbpA molecule (residues 1–309) and 238 water molecules in the asymmetric unit. Data collection and refinement statistics are summarized in Appendix A. The Ramachandran plot of the apo VmFbpA structure shows that 99.35% and 0.65% of the residues are in the most favored and allowed regions, respectively. Second, the binding and inhibiting models were simulated in silico by docking analysis using Autodock vina. Three areas were selected as the potential docking regions for VmFbpA binding, namely C-Lobe, N-Lobe, and Fe^3+^ binding site (Appendix A). The Δ*G* value for each model docking at the C-Lobe and N-Lobe was 0.0 kcal/mol, whereas the Δ*G* value for the best model docking at the Fe^3+^ binding site was ≈−7.0 kcal/mol, indicating that RA binds VmFbpA at the Fe^3+^ binding site rather than C-Lobe and N-Lobe (Appendix A).

Inside the Fe^3+^ binding site, a total of nine models was presented by AutoDock (version Vina 1.1.2) (Figure 7, Appendix A). In Model 1, six hydrogen bonds were formed between RA and VmFbpA. Amino acid residues R9, G140, N193, and V259 were involved in the interaction (Figure 7a). In Model 2, eight hydrogen bonds were formed between RA and VmFbpA. Amino acid residues R9, G34, D35, T36, Q58, and Y196 were involved in the interaction (Figure 7b). In Model 3, seven hydrogen bonds were formed between RA and VmFbpA. Amino acid residues R9, Q58, G140, N193, Y195, and Y196 were involved in the interaction (Figure 7c). In Model 4, six hydrogen bonds were formed between RA and VmFbpA. Amino acid residues R9, N138, N193, and Y196 were involved in the interaction (Figure 7d). In Model 5, four hydrogen bonds were formed between RA and VmFbpA. Amino acid residues R9, Q58, and R100 were involved in the interaction (Figure 7e). In Model 6, six hydrogen bonds were formed between RA and VmFbpA. Amino acid residues R9, R100, Y196, and E262 were involved in the interaction (Figure 7f). In Model 7, six hydrogen bonds were formed between RA and VmFbpA. Amino acid residues Q58, Y195, and R199 were involved in the interaction (Figure 7g). In Model 8, four hydrogen bonds were formed between RA and VmFbpA. Amino acid residues Q58, N193, Y195, and E262 were involved in the interaction (Figure 7h). In Model 9, four hydrogen bonds were formed between RA and VmFbpA. Amino acid residues R9, Q58, N193, and E262 were involved in the interaction (Figure 7i).

### 2.9. Inhibition by RA and SC of the Growth of V. vulnificus and V. parahaemolyticus

To assess the bacteriostatic effects of RA and SC on other *Vibrio* species, *V*. *parahaemolyticus* (which causes the most *Vibrio* infections in the US) and *V*. *vulnificus* (which is the most life-threatening non-cholera *Vibrio* bacterium), were used. *V*. *vulnificus* was almost completely inhibited by 100 mM RA and 100 mM SC, whereas *V*. *parahaemolyticus* was resistant to the same condition (Figure 8a). However, at 1000 mM RA, the growth of both *V*. *vulnificus* and *V*. *parahaemolyticus* was almost completely inhibited (Figure 8b).

## 3. Discussion

### 3.1. Screening and Identification of RA as a VmFbpA Inhibitor

Fe concentration was quantified via a VmFbpA inhibitor screening assay using ICP-MS. Rosemary and cinnamon extracts decreased Fe concentrations to almost half of that in the Holo group (Fe^3+^-VmFbpA, positive control) (Figure 1a). UV-vis spectral analysis indicated a changed spectral pattern in samples treated with rosemary and cinnamon (Figure 1c,e). A significant peak shift (from ≈412 to ≈380 nm) in the UV-vis spectra was observed in samples treated with rosemary extracts (Figure 1e) but not in those treated with cinnamon extracts (Figure 1c). This indicates that the chemicals in the rosemary extracts interact with or bind to VmFbpA. By contrast, the interaction between cinnamon extracts and VmFbpA may be nonspecific. 

In rosemary, the most frequently reported functional chemical is RA, which is the main active phenolic compound and has antibacterial activity [16]. Consequently, a spectral analysis of apo VmFbpA supplemented with RA was performed (Figure 1f). The same peak shift (from ≈412 to ≈380 nm) as for rosemary extracts was observed. Therefore, the functional chemical that specifically interacts with VmFbpA was RA.

In the spectra of RA only (Figure 1g), no specific peak at ≈380 nm was observed. This demonstrated that the peak at ≈380 nm was a result of binding between VmFbpA and RA but not RA itself (Figure 1f,g).

### 3.2. Interaction between RA and VmFbpA

The ITC results demonstrated that the interaction between RA and apo VmFbpA takes place at a single site 1:1 binding with a *K*_D_ value of 8 ± 1 μM (Figure 3a), which could be presented as below:RA+VmFbpA ⇌RA·VmFbpA (Reaction I).

Based on the thermodynamic parameters obtained from the ITC results, Δ*G* for Reaction I at 298 K is −5 ± 2 kcal/mol, indicating that Reaction I can progress spontaneously without external energy (Figure 3d). Furthermore, the ∆*H* and −T∆*S* for Reaction I were −15 ± 9 kcal/mol and 11 ± 8 kcal/mol, respectively, indicating that Reaction I was driven by enthalpy but not entropy and hydrogen bonds were the main interactions [17,18] between RA and VmFbpA (Figure 3d).

The interaction between RA and Fe^3+^-VmFbpA was more complicated (Figure 3b). In the first four titrations, the heat change (μcal/s) experienced a decrease to a negative value (an exothermic reaction), which was followed by an increase to a positive value (an endothermic reaction), so that the net heat change was zero. An exothermic reaction was observed from the fifth to the eleventh titration, after which VmFbpA became saturated. The fact that an exothermic reaction followed by an endothermic reaction was observed in the first four titrations is a result of complex interactions among Fe^3+^, VmFbpA, and RA. These interactions could include binding between RA and VmFbpA (exothermic reaction), binding between Fe^3+^ and VmFbpA (endothermic reaction) (Figure 3c) [19], and reduction of Fe^3+^ to Fe^2+^ by RA (Figure 2e). A significant heat difference was observed only when the molar ratio of RA: Fe^3+^-VmFbpA was higher than 2:1, indicating that RA binds to VmFbpA competitively at the Fe^3+^ binding site and Fe^3+^-VmFbpA has some resistance to RA at low concentrations.

Docking results showed that RA binds VmFbpA at the Fe^3+^ binding site (ΔG ≈ −7.0 kcal/mol) (Appendix A) rather than the C-Lobe and N-Lobe (ΔG ≈ 0.0 kcal/mol). The in silico theoretical *K*_D_ value between RA and apo VmFbpA is ≈7.1 μM at room temperature, which is consistent with the in vitro ITC results. In the docking model, RA formed many hydrogen bonds with VmFbpA (Figure 7), indicating that the interaction between RA and VmFbpA is driven by enthalpy but not entropy, as described in the analysis of ITC results.

### 3.3. Interaction between RA and Fe^3+^

The net spectra of RA in the presence of Fe^3+^ indicated that RA interacts with Fe^3+^ (Appendix A), and the iron reduction experiments showed that RA can reduce Fe^3+^ to Fe^2+^ (Figure 2e,f). RA is an effective antioxidant, showing DPPH scavenging [20], ABTS(+) scavenging [21], and other activities. However, our results suggest that the antioxidant activity of RA is strong enough to reduce Fe^3+^ to Fe^2+^, which is consistent with a report that RA and its derivatives can reduce the Fe^3+^ in ferric 2,4,6-tris(2-pyridyl)-*s*-triazine (TPTZ) to Fe^2+^ to a colored product [22].

Therefore, Fe^3+^ release from VmFbpA by RA was in part caused by the competitive inhibition of VmFbpA and by the reduction of Fe^3+^ to Fe^2+^. The regression curve showed that 1 μM RA can reduce ≈2.44 μM Fe^3+^ to Fe^2+^ (Figure 2f), indicating that the stoichiometry of the reaction between RA and Fe^3+^ is approximately 1:2 and one RA molecule can reduce about two Fe^3+^ to Fe^2+^ (Figure 2f).

### 3.4. Inhibition of V. metschnikovii Growth by RA

The growth of *V*. *metschnikovii* was significantly inhibited by RA in a dose-dependent manner (Figure 4a). This is probably because the iron-capturing protein, VmFbpA, was inhibited by RA (Figure 2c,d), and Fe utilization in *V*. *metschnikovii* is significantly lowered (Figure 5b). The recovery experiments showed that the bactericidal activity of RA against *V*. *metschnikovii* resulted from the restriction of iron uptake, because the growth of *V*. *metschnikovii* was recovered after adding 5 μmol FeCl_3_ (Figure 5c,d).

*E*. *coli* is the most abundant bacterium in the human colon. Therefore, we evaluated the growth of *E*. *coli* with the supplementation of RA. *E*. *coli* was not significantly inhibited by RA (Figure 4c,d). Similarly, it has been reported that *E*. *coli* is more resistant to rosemary extracts and RA than other Gram-negative bacteria [16]. Moreover, *E*. *coli* can use a wide range of iron sources, for instance, ferric ammonium citrate, hematin hydrochloride, lysed guinea-pig red cells, crystalline human hemoglobin [23], and ferrous ion [24]. Therefore, even if FbpA, the Fe^3+^ capturing protein, is inhibited, *E*. *coli* could use other iron sources for growth (Table 1).

Additionally, it is important to note that the pH of the NB medium did not change significantly after adding RA and SC (Appendix A). Therefore, it is not the pH change that affects the growth of bacteria.

### 3.5. Effect of RA Combined with SC on Bacterial Growth

RA (1000 μM) inhibited the growth of *V*. *metschnikovii* to ≈65% (Figure 4a,b), and the inhibition was enhanced to >90% by adding SC (Figure 6a,b). However, RA did not significantly inhibit *E*. *coli* growth (Figure 4c,d), nor did the combination of RA and SC (Figure 6c,d). These results indicate that *E*. *coli* uses a variety of iron sources and is resistant to RA and SC, but *V*. *metschnikovii* is more vulnerable to RA in the presence of SC. *E*. *coli* and *V*. *metschnikovii* have an FbpBC/A uptake system to take up Fe^3+^, which is a common means of absorbing iron (Table 1). Under anaerobic, low pH, or reducing conditions, iron is mainly present as Fe^2+^ [25]. *E*. *coli* has evolved diverse transporter systems (FeoABC [27], EfeOUB [24], and MntH [28]) to absorb Fe^2+^, whereas *V*. *metschnikovii* has only an FeoABC system (Table 1).

Interestingly, *E*. *coli* responds to iron restriction by producing citrate [30,31] and has an Fec(CD)E/A transporter system for ferric citrate uptake [29], which is absent in *V*. *metschnikovii* (Table 1). Furthermore, FecB, the periplasmic subunit of the Fec(CD)E/B system, binds a variety of Fe-citrate complexes (citrate, Fe^3+^-Cit, [Fe_2_(Cit_2_)]^2-^, and [Fe(Cit)_2_]^5-^) as well as other citrate complexes (Ga^3+^, Al^3+^, Sc^3+^, In^3+^, and Mg^2+^) with a *K*_d_ value in the micromolar range [29]. When FbpAs in *E*. *coli* and *V*. *metschnikovii* were inhibited by 1000 μM RA, Fe^3+^ was reduced to Fe^2+^. Subsequently, adding citrate chelates free Fe^2+^ into Fe^2+^-citrate complexes. These Fe^2+^-citrate complexes can be taken up by *E*. *coli* via the Fec(CD)E/B transporter system, whereas the Fe^2+^-citrate complexes are not bioavailable for *V*. *metschnikovii*. This could be why *E*. *coli* is more resistant to RA and the combination of RA and SC than *V*. *metschnikovii*. Therefore, RA combined with SC could inhibit *V*. *metschnikovii* without deleterious effects on indigenous gastrointestinal bacteria.

Moreover, the following reasons may explain why SC did not affect the growth of *V*. *metschnikovii*. SC chelates Fe^3+^ to Fe^3+^-citrate complexes, but *V*. *metschnikovii* has a siderophore-based CeuBC/A transporter that is responsible for Fe^3+^-enterochelin uptake (Table 1). *V*. *metschnikovii* releases the siderophore enterochelin to absorb Fe^3+^ under iron-restricted conditions. Enterochelin is the strongest siderophore and binds to Fe^3+^ with extremely strong affinity (*K*a = 10^52^ M^−1^) [32]. This value is substantially higher than even several synthetic metal chelators, such as EDTA (*K*a = 10^25^ M^−1^) [33]. Therefore, without RA, Fe^3+^ was not reduced to Fe^2+^. If Fe^3+^ is chelated by citrate, it can be reclaimed by enterochelin. This explains why the growth of *V*. *metschnikovii* was not affected by SC alone.

Considering the future potential applications of VmFbpA inhibitors, we conducted an alignment of VmFbpA to other FbpAs from pathogenic *Vibrio* species. Since we focused on non-cholera *Vibrio* infections in this study, *V*. *vulnificus* and *V*. *parahaemolyticus* were selected for the further antibacterial experiments. The results showed that FbpA is a highly conserved protein with a sequence identity of ≈31% (Appendix A). Bacteriostatic experiments (Figure 5) on *V*. *vulnificus* and *V*. *parahaemolyticus* indicated that RA combined with SC can also inhibit the growth of these infectious *Vibrio* by limiting Fe^3+^ uptake. Based on the knowledge iron uptake proteins in *Vibrio* bacteria, the RA/SC would be further effective in suppressing the growth of *V. cholerae*, although it needs to be confirmed.

Additionally, the results from a pilot experiment performed in Japan several decades ago support our experimental data [34]. The traditional Japanese food “umeboshi” is made with shiso (perilla; *Perilla frutescens* var. *crispa*) and ume (salted Japanese plum; *Prunus mume* Sieb. et Zucc.), which contain RA and SC, respectively. Due to its antibacterial properties, umeboshi was traditionally used as a medicine in the middle and early modern ages and is currently used as a natural preservative to prevent the spoilage of items in lunch boxes and rice balls. In the 19th and early 20th centuries, umeboshi was found to be an effective treatment for cholera and was used as an anti-cholera medicine in Japan. Tetsumoto (1934) found that umeboshi exhibited strong bactericidal activity against *V. cholerae*, as compared to citric acid, which has a similar pH [34]. Our data presented here suggest that RA and SC are among the anti-cholera components in umeboshi. Here, we propose that the anti-cholera activity of umeboshi could be exerted mainly or partly by inhibiting the iron uptake by *V. cholerae*, although it needs to be confirmed by further experiments.

## 4. Materials and Methods

### 4.1. Chemicals and Materials

RA (98% pure, CAS: 20283–92–5) was purchased from Combi-Blocks (San Diego, CA, USA). All other reagents used in this study were purchased from Fujifilm Wako Pure Chemical Co., Ltd. (Osaka, Japan) unless otherwise stated.

Twenty spices, including Sanshō (Zanthoxylum piperitum), oregano (Origanum vulgare), thyme (Thymus vulgaris), allspice (Pimenta dioica), rosemary (Salvia rosmarinus), basil (Ocimum basilicum), coriander (Coriandrum sativum), parsley (Petroselinum crispum), cumin (Cuminum cyminum), cumin seeds, ginger (Zingiber officinale), cinnamon (Cinnamomum verum), paprika (Capsicum annuum), garam masala, garlic (Allium sativum), nutmeg (Myristica fragrans), turmeric (Curcuma longa), chili pepper (Capsicum annuum), black pepper (Piper nigrum), and white pepper (Piper nigrum) used were the products of S&B Foods (Tokyo, Japan).

*Vibrio* species (*V. metschnikovii*, *V. vulnificus*, and *V. parahaemolyticus*) were obtained from Japan Collection of Microorganisms, RIKEN (Ibaraki, Japan) and were handled in a safety cabinet to avoid contamination. All *E. coli* strains (BL21(DE3) and KRX) were purchased from Novagen (Madison, WI, USA) and Promega (Madison, WI, USA), respectively, and were handled in a clean bench to avoid contamination. 

### 4.2. Aqueous Extracts of Spices

A total of 0.50 g (±0.01) of each spice was ground into powder, extracted in 20 mL boiling Milli-Q (Milli-Q Advantage), and incubated in a water bath at 95 °C for 3 h. The insoluble fraction was removed using a 0.45 μm polyethersulfone (PES) filter. The filtrate was evaporated to powder and stored at −30 °C.

### 4.3. Expression and Purification of 6× His-tev-VmFbpA

The gene encoding VmFbpA (WP_004394209.1) without the signal peptide ‘MQKKLLSTVALVSSLITAPLLHA’ was cloned into a modified pET-28a vector, where the thrombin cleavage site had been replaced by a TEV cleavage site, and overexpressed in *E*. *coli* BL21(DE3). Large-scale overexpression of 6× His-tev-VmFbpA was performed at 30 °C in 2 L of LB medium with isopropyl β-D-1-thiogalactopyranoside (final concentration 0.001 mM) at an OD_600_ of about 0.6. After the induction, the cells were cultivated at 30 °C for 20 h and harvested by centrifugation at 4000× *g* and 4 °C for 20 min. The cells were suspended in lysis buffer (20 mM Tris-HCl (pH 8.0), 100 mM NaCl, 10 mM imidazole, 0.50 mg/mL lysozyme, and 1.2 U/mL DNase I). Subsequently, the cells were disrupted by sonication using 70% amplitude, 0.5 s on/off, for 7 min, and the resulting cell lysate was centrifuged at 40,000× *g* and 4 °C for 30 min. The supernatant was loaded onto a Ni-nitrilotriacetic acid (Ni-NTA) column (bed volume 2 mL), washed and eluted sequentially by wash buffer 1 (20 mM Tris-HCl (pH 8.0), 100 mM NaCl, and 10 mM imidazole), wash buffer 2 (wash buffer 1 supplemented with 20 mM imidazole), elution buffer 1 (wash buffer 1 with 50 mM imidazole), elution buffer 2 (wash buffer 1 with 100 mM imidazole), elution buffer 3 (wash buffer 1 with 150 mM imidazole), and elution buffer 4 (wash buffer 1 with 250 mM imidazole). Based on the sodium dodecyl sulfate-polyacrylamide gel electrophoresis (SDS-PAGE) data, all fractions containing 6× His-tev-VmFbpA were collected and stored at 4 °C before use.

### 4.4. Preparation of 6× His-tag-free VmFbpA

A total of 6× His-tev-VmFbpA were dialyzed overnight against lysis buffer at 4 °C to lower the imidazole concentration. The 6× His-tag was cleaved by adding 6× His-TEV protease (homemade, 10 units/mg protein). Then, the reaction mixture was reloaded on the Ni-NTA column and the eluent, which contained mainly 6× His-tag-free VmFbpA, and was collected and concentrated with a Vivaspin 20 (MWCO 10 K, Sartorius) to a protein concentration of approximately 1.5 mg/mL. Then, the sample was loaded onto a Superdex 200 10/300 GL column (GE Healthcare) and separated using gel filtration buffer A (20 mM Tris-HCl (pH 8.0) and 100 mM NaCl) using an ÄKTA purifier (GE Healthcare) for the final purification. Protein samples were adjusted to a final protein concentration of 0.3 mM by ultrafiltration using a Vivaspin 20 (MWCO 10 K, Sartorius) and stored at 4 °C.

### 4.5. Preparation of Fe^3+^-VmFbpA and Apo VmFbpA

To form Fe^3+^-bound VmFbpA, ferric chloride hexahydrate (FeCl_3_·6H_2_O) was added to purified VmFbpA to a final concentration of 4.2 mM. Next, EDTA at a final concentration of 10 mM was added to form apo VmFbpA. The samples were incubated with 50 mM sodium bicarbonate (NaHCO_3_) at 4 °C for 20 h and then concentrated to a protein concentration of approximately 0.7 mg/mL. Each sample was loaded onto a Superdex 200 10/300 GL column and separated using gel filtration buffer (50 mM Tris-HCl (pH 8.0), 50 mM NaHCO_3_, and 150 mM NaCl). Fractions were collected according to the peak measured at 280 nm. The samples were adjusted to a final protein concentration of 0.1 mM by ultrafiltration using a Vivaspin 20 (MWCO 10 K) and stored at 4 °C.

### 4.6. Six × His-based VmFbpA Inhibition Assay (Pull-Down Assay)

Two-hundred microliters of 0.3 mM apo 6× His-tev-VmFbpA were loaded onto a 40 μL Ni-NTA slurry resin (bed volume: 20 μL) in a Micro Bio-Spin column (Bio-Rad) and incubated at 4 °C for 30 min with 3.0 mM FeCl_3_. The column was extensively washed with washing buffer (50 mM Tris-HCl (pH 8.0), 50 mM NaHCO_3_, 150 mM NaCl, and 20 mM imidazole). Spice extract or RA (200 μL) was loaded onto the column and incubated at 4 °C for 60 min. The solvent for spice extracts and RA was 50 mM Tris-HCl (pH 8.0), 50 mM NaHCO_3_, and 150 mM NaCl. The column was subsequently washed extensively with washing buffer (50 mM Tris-HCl (pH 8.0), 50 mM NaHCO_3_, 150 mM NaCl, and 20 mM imidazole) and eluted with 300 µL elution buffer (50 mM Tris-HCl (pH 8.0), 50 mM NaHCO_3_, 150 mM NaCl, and 250 mM imidazole). The Fe concentration of eluent was determined by ICP-MS as reported previously [19] and adjusted to the protein concentration (measured using a Pierce 660 nm Protein Assay, Thermo Fisher Scientific) in the eluent. Samples without FeCl_3_ were used as the negative control group, and samples lacking rosemary extract or RA as the positive control group. Unless indicated, all buffers contained 0.3% dimethyl sulfoxide (DMSO) [35] to solubilize RA.

### 4.7. Determination of the 50% Inhibitory Concentration

Rosemary extracts and RA were used to evaluate VmFbpA inhibitory activity. Two-hundred microliters of rosemary extracts (1.56 to 25.0 mg/mL) and RA (2.34 μg/mL to 2.40 mg/mL), with a two-fold dilution gradient were performed. Apo VmFbpA was used as the negative control and Fe^3+^-VmFbpA lacking RA was used as the positive control. The Fe concentration of the eluents was determined by ICP-MS as reported previously [19]. The inhibitory rates were calculated as below:
$$\mathrm{Inhibitory}\;\mathrm{rates}\;\left(\%\right)=\left[1-\frac{\left(\mathrm{Conc}.{\mathrm{Fe}}_{\mathrm{sample}}-\mathrm{Conc}.{\mathrm{Fe}}_{\mathrm{apo}}\right)}{\left(\mathrm{Conc}.{\mathrm{Fe}}_{\mathrm{holo}}-\mathrm{Conc}.{\mathrm{Fe}}_{\mathrm{apo}}\right)}\right]\times 100\%$$

The inhibitory rate was plotted in a scatter plot as a function of RA concentration and fitted by a logarithmic function model. The 50% inhibitory concentration (IC_50_) was calculated by reading the RA concentration at the inhibitory rate is 50%.

### 4.8. Spectral Analysis of VmFbpA Interacting with Rosemary Extracts and RA

Rosemary extracts (15 to 0.9 mg/mL, with a two-fold dilution gradient) and 200 µL 0.3 mM apo 6× His-tev-VmFbpA with 3 mM FeCl_3_ were used in a pull-down assay to assess the interaction between Fe^3+^-VmFbpA and RA.

RA (2.4 to 0.15 mg/mL, with a two-fold dilution gradient) and 200 µL 0.3 mM apo 6× His-tev-VmFbpA without FeCl_3_ were used in the pull-down assay to assess the interaction between apo VmFbpA and RA.

The absorbance spectra of the eluents at 300–700 nm were measured using a UV–vis spectrophotometer (JASCO V-630) at room temperature. Samples without FeCl_3_ and RA were used as the negative controls and samples lacking RA were used as the positive control. Samples without preloading of apo 6× His-tev-VmFbpA were used to determine the background.

### 4.9. Spectral Analysis of RA Interacting with Fe^3+^

Solutions of 20 μM RA supplemented with 0, 10, 20, 30, and 40 μM FeCl_3_ were used for UV-vis spectra analysis. The absorbance at 200–700 nm was measured using a UV-vis spectrophotometer (Jasco V-630) at room temperature. Solutions of 0, 10, 20, 30, and 40 μM FeCl_3_ without 20 μM RA were used as the controls. The buffers did not contain DMSO.

### 4.10. Analysis of RA on Fe^3+^ Reduction

Solutions of 40 μM FeCl_3_ supplemented with 0, 0.625, 1.25, 5, 10, 20, and 40 μM RA were used for analysis. First, 0.25% o-phenanthroline (2 mL) was added to each sample, and the pH was adjusted to 3.5 with hydrochloric acid (HCl). Milli-Q water was added to a final volume of 25 mL. After 1 h incubation, the absorbance at 510 nm was measured using a microplate spectrophotometer (Benchmark Plus, Bio-Rad). The concentration of Fe^2+^ was quantified using ferrous sulfate heptahydrate (FeSO_4_·7H_2_O) as the standard. The buffers did not contain DMSO.

### 4.11. ITC Analysis

The 6× His-free apo/Fe^3+^ VmFbpA samples were dialyzed overnight at 4 °C against dialysis buffer (20 mM Tris-HCl (pH 8.0), 25 mM NaHCO_3_, and 50 mM NaCl, 20 mM imidazole, 0.3% DMSO). ITC was performed using MicroCal iTC200 (Malvern). RA (0.5 mM) or Fe^3+^ (0.1 mM) diluted in dialysis buffer was injected 20 times by a motor-driven syringe into 200 µL 0.01 mM Apo VmFbpA. The first injection volume was 0.1 μL, and the subsequent injection volumes were 2.0 μL. The solution in the titration cell was stirred at 750 rpm throughout the experiment. In addition, titration of RA (0.5 mM) into Fe^3+^-VmFbpA (0.1 mM) was performed as described above. The reference cell of the microcalorimeter was filled with 200 µL of the corresponding dialysis buffer. The titrations were performed at room temperature. The data were analyzed by MicroCal LLC iTC 200 for Windows (Malvern).

### 4.12. Crystallization and X-ray Diffraction Data Collection

Initial crystallization screening was performed in 96-well Violamo Protein Crystallization Plates (As One) using commercially available kits, namely Crystal Screen HT, Index HT (Hampton Research), and Wizard I and II (Emerald BioSystems) at 293 K. A crystallization drop was prepared by mixing 0.5 µL protein solution and 0.5 µL reservoir solution and was equilibrated against 40 µL reservoir solution. After two-dimensional grid optimization of the crystallization conditions (pH versus precipitant concentration), they were further optimized by the sitting-drop vapor-diffusion method in 24-well Cryschem Plates (Hampton Research). A crystallization drop was prepared by mixing 1.0 µL protein solution and 1.0 µL reservoir solution and was equilibrated against 500 µL reservoir solution.

Crystals of apo VmFbpA were obtained using a reservoir composition of 0.25 M ammonium tartrate dibasic, 25% PEG3350, 100 mM Tris-HCl (pH 7.0). Each crystal was picked up in a mounting loop and cooled in a cold nitrogen-gas stream using a cryoprotectant consisting of 0.25 M ammonium tartrate dibasic, 25% PEG3350, 100 mM Tris-HCl (pH 7.0), and 20% (*v*/*v*) glycerol. X-ray diffraction experiments for apo VmFbpA crystals were performed on beamlines BL44XU at SPring-8 (Hyogo, Japan) and AR-NE3A at Photon Factory (Ibaraki, Japan). The final data set was collected from an apo VmFbpA crystal at a wavelength of 0.899995 Å, a crystal-to-detector distance of 319.90 mm, an oscillation angle of 0.1°, and an exposure time of 0.1 s per image using an EIGER 16M detector at BL44XU at SPring-8. The structure solution and refinement were described in the Appendix A.

### 4.13. Docking Simulation

The crystal structure of apo VmFpbA solved in this study was used as the template. Docking simulation analysis was performed on a Linux (CentOS 7.0) platform using AutoDock (version Vina 1.1.2) [36]. Three areas were selected (N-terminus, C-terminus, and the Fe^3+^-binding site) (Appendix A) as the docking cavities for the grid box to cover. The 3D model of RA (Appendix A) was created by ChemBioDraw Ultra (version 13.0.0.3015) and optimized by AutoDockTools (version 1.5.4). The default mode was used.

### 4.14. Inhibition by RA of V. metschnikovii Growth

The glycerol stock of *V*. *metschnikovii* was prepared by culturing the bacteria in NB liquid medium (meat extract 3.0 g/L, meat peptone 5.0 g/L, NaCl 10.0 g/L, at pH 7.0) and flash freezing in liquid nitrogen after adding 50% glycerol (1:1, *v*/*v*) at the OD_600_ of 0.6. Next, 10 µL *V*. *metschnikovii* cell from the glycerol stock were added to 10 mL NB medium containing 0.3% DMSO and 1000, 500, 100, 50, 25, and 0 μM RA. The culture was performed at 37 °C. The optical density at 600 nm was monitored every h (0–10 h) by a microplate spectrophotometer (Benchmark Plus, Bio-Rad).

### 4.15. Inhibition of E. coli Growth by RA

A glycerol stock of *E*. *coli* (BL21(DE3) or KRX) was prepared by culturing the bacteria in NB medium and flash freezing in liquid nitrogen after adding 50% glycerol (1:1, *v*/*v*) to an OD_600_ of 0.6. *E*. *coli* (BL21(DE3) or KRX) (10 μL) from the glycerol stock was added to 10 mL NB medium containing 0.3% DMSO and 1000, 500, 100, 50, 25, and 0 μM RA. The culture was performed at 37 °C. The optical density at 600 nm was monitored every hour (0–10 h) by a microplate spectrophotometer (Benchmark Plus, Bio-Rad).

### 4.16. Measurement of Iron Utilization by V. metschnikovii

*V*. *metschnikovii* (10 μL) from the glycerol stock were added to 10 mL NB medium containing 0.3% DMSO and 1000, 500, 100, 50, 25, and 0 μM RA. The culture was performed at 37 °C for 7 h followed by centrifugation at 40,000 g for 15 min. The weight of precipitate (*V*. *metschnikovii*) was measured using a precision balance. The remaining iron in supernatant was measured using ICP-MS as reported previously [19]. The Fe utilization ratio was calculated as:$$\mathrm{Fe}\;\mathrm{utilization}\;\mathrm{ratio}\;\left(\mu \mathrm{mol}\cdot L^{-1}\cdot g^{-1}\right)=\frac{{\mathrm{Fe}}_{\mathrm{Conc},\mathrm{BK}}-{\mathrm{Fe}}_{\mathrm{Conc},\mathrm{Remian}}}{m_{vibrio}\times V}$$
where Fe_Conc,BK_ is the original iron concentration in NB medium, Fe_Conc,Remain_ is the remaining iron concentration after culture of *V*. *metschnikovii*, m*_vibrio_* is the weight of *V*. *metschnikovii* after culture, and V is the volume of NB medium.

### 4.17. Recovery of V. metschnikovii by Supplementation of Fe^3+^

*V*. *metschnikovii* (10 μL) from the glycerol stock was added to 10 mL NB medium containing 1000 μM RA with 10 and 50 μmol FeCl_3_, respectively. The positive control group was *V*. *metschnikovii* without supplementation of RA and FeCl_3_. The negative control group was *V*. *metschnikovii* without FeCl_3_. Culture was performed at 37 °C. The optical density at 600 nm was monitored every hour (0–10 h) by a microplate spectrophotometer (Benchmark Plus, Bio-Rad).

### 4.18. Inhibition by RA and SC of the Growth of V. metschnikovii

*V*. *metschnikovii* (10 μL) from the glycerol stock was added to 10 mL of NB medium supplemented with RA and SC. Culture was performed at 37 °C for 7 h. Optical density at 600 nm was measured using a microplate spectrophotometer (Benchmark Plus, Bio-Rad).

### 4.19. Inhibition by RA and SC of the Growth of E. coli

*E*. *coli* (BL21(DE3) or KRX) (10 μL) from the glycerol stock was added to 10 mL NB medium supplemented with RA and SC. Culture was performed at 37 °C for 7 h. The optical density at 600 nm was measured using a microplate spectrophotometer (Benchmark Plus, Bio-Rad).

### 4.20. Inhibition by RA and SC of the Growth of V. vulnificus and V. parahaemolyticus

Glycerol stocks of *V*. *vulnificus* and *V*. *parahaemolyticus* were prepared as for *V*. *metschnikovii*. Cells (10 μL) from the glycerol stock were added to 10 mL NB medium supplemented with RA and SC. Culture was performed at 37 °C for 7 h. The optical density at 600 nm was measured using a microplate spectrophotometer (Benchmark Plus, Bio-Rad).

### 4.21. Statistical Analysis

The statistical significance was analyzed by one-way ANOVA followed by a Tukey HSD (honestly significant difference) post hoc test using R program (Version 3.6.0). *p* ≤ 0.05 was considered to be statistically significant.

## 5. Conclusions

The advent of multidrug resistance among pathogenic bacteria is imperiling the utility of antibiotics. The inhibition of FbpA and iron uptake shows promise as an antibacterial. In this study, we have revealed the molecular mechanism of anti-*Vibrio* activity of RA. The growths of *V. metschnikovii*, *V*. *vulnificus*, and *V*. *parahaemolyticus* were inhibited significantly by RA and further inhibited by the supplementation of SC. The combination of RA and SC has high potential to be applied as a food preservative and bacteriostatic agent in the future. Moreover, such new bacteriostatic agents will reduce the excessive and inappropriate use of antibiotics in livestock, aquaculture, and plant agriculture as well as human and veterinary medicine. Further studies should evaluate the utility of RA/SC in various areas.

## Figures and Tables

**Figure 1 ijms-22-13010-f001:**
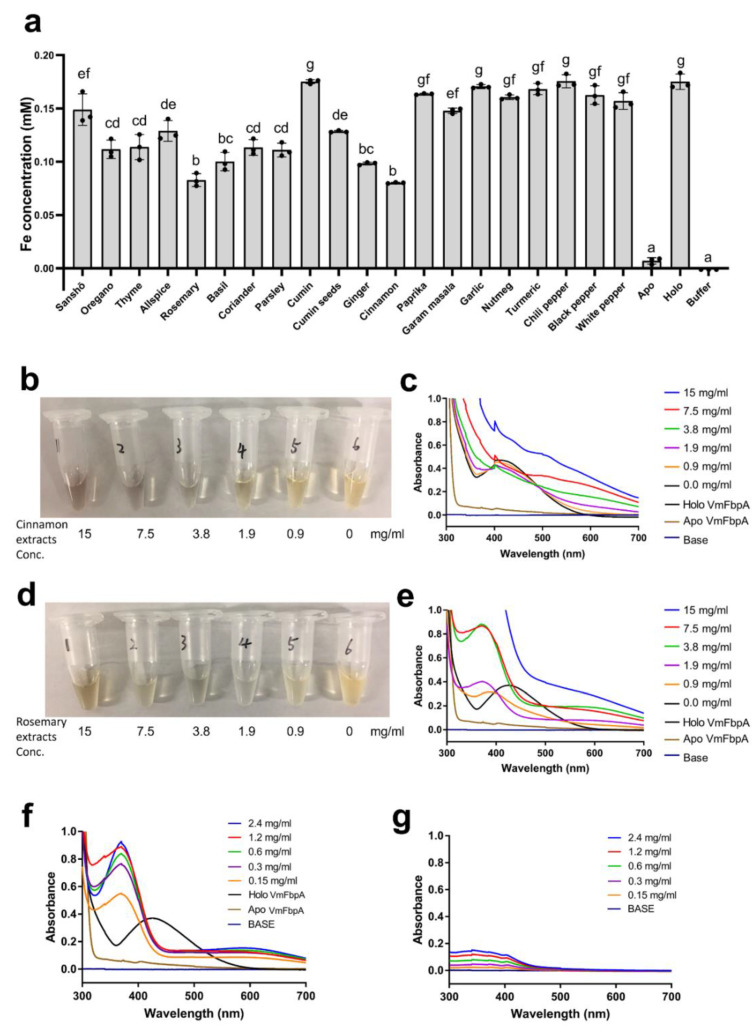
(**a**) Screening of VmFbpA inhibitors for Fe^3+^ binding. The gray bar represents the remaining Fe concentration after treatment with the extracts of 20 spices; 10 mM EDTA was used as the negative control. In addition, apo VmFbpA (Apo) and Fe^3+^-VmFbpA (Holo) were also measured for comparison. Buffer represents the gel filtration buffer (50 mM Tris-HCl (pH 8.0), 50 mM NaHCO_3_, and 150 mM NaCl) used in VmFbpA purification. Data are means ± SD. Means with the same letter are not significantly different from each other (*p* < 0.05) (**b**) Eluents after inhibition assays using cinnamon extracts. (**c**) Absorbance spectra (300–700 nm) of the respective eluents shown in (**b**). (**d**) Eluents after inhibition assays using rosemary extracts. (**e**) Absorbance spectra (300–700 nm) of the respective eluents shown in (**c**). Spectra of apo and Fe^3+^-VmFbpA (controls). (**f**) Absorbance spectra (300–700 nm) of RA binding to apo VmFbpA. (**g**) Absorbance spectra (300–700 nm) of RA only. Base is elution buffer containing 50 mM Tris-HCl (pH 8.0), 50 mM NaHCO_3_, 150 mM NaCl, and 250 mM imidazole.

**Figure 2 ijms-22-13010-f002:**
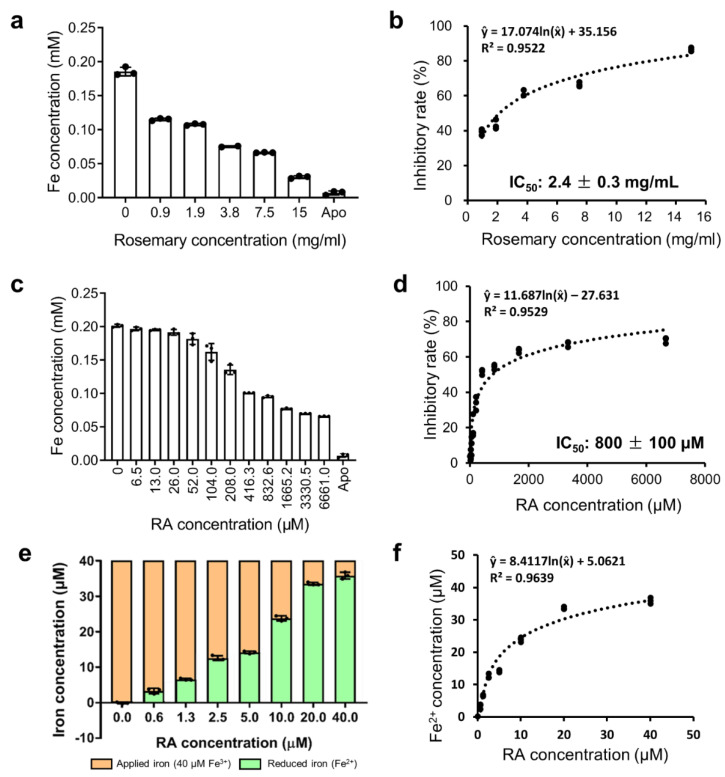
(**a**) The remaining Fe concentrations in the eluents after inhibition assays using rosemary extracts. (**b**) Inhibitory rates of rosemary extracts. A nonlinear regression model was performed to fit the concentration (rosemary)—response (inhibitory rate) data. (**c**) The remaining Fe concentrations in the eluents after inhibition assay using RA. (**d**) Inhibitory rates of RA. A nonlinear regression model was performed to fit the concentration (RA)—response (inhibitory rate) data. (**e**) The reduced Fe^2+^ concentration after adding RA (0, 0.6, 1.3, 2.5, 5, 10, 20, and 40 μM). Orange bar, applied Fe^3+^ (40 μM). Green bar, reduced Fe^2+^ from Fe^3+^ by RA. (**f**) A nonlinear regression curve fitting of the concentration (RA)—response (Fe^2+^) data.

**Figure 3 ijms-22-13010-f003:**
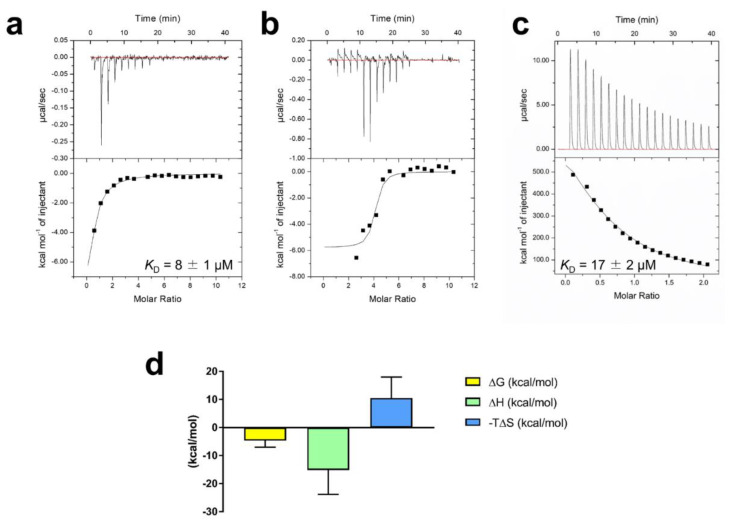
ITC analysis of RA binding to VmFbpA at 298 K. (**a**) Binding between 10 μM apo VmFbpA and 500 μM RA. (**b**) Binding between 10 μM Fe^3+^-VmFbpA and 500 μM RA. (**c**) Binding between 10 μM apo VmFbpA and 100 μM Fe^3+^. (**d**) Thermodynamic parameters for the binding of RA to apo VmFbpA calculated according to the data in (**a**). The free energy of binding (Δ*G*, light yellow bar) is the sum of the enthalpic (Δ*H*, light green bar) and entropic (−TΔ*S*, light blue bar) terms.

**Figure 4 ijms-22-13010-f004:**
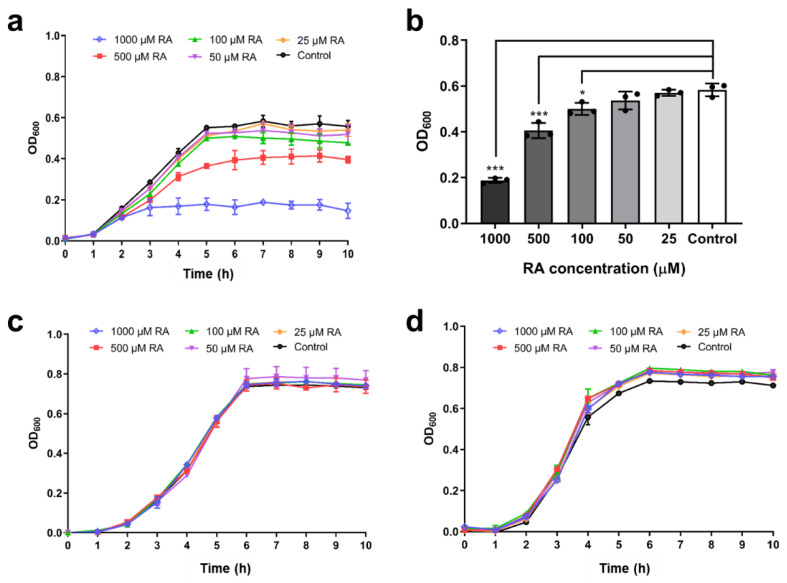
Antibacterial effect of RA. (**a**) Inhibitory effects of RA on *V. metschnikovii* growth from 0 to 10 h. (**b**) Inhibitory effects of RA on *V. metschnikovii* growth at 7 h. (**c**) Inhibitory effects of RA on *E. coli* (KRX) growth from 0 to 10 h. (**d**) Inhibitory effects of RA on *E. coli* (BL21(DE3)) growth from 0 to 10 h. RA was used at 0, 25, 50, 100, 500, and 1000 μM. * *p* < 0.05, *** *p* < 0.001.

**Figure 5 ijms-22-13010-f005:**
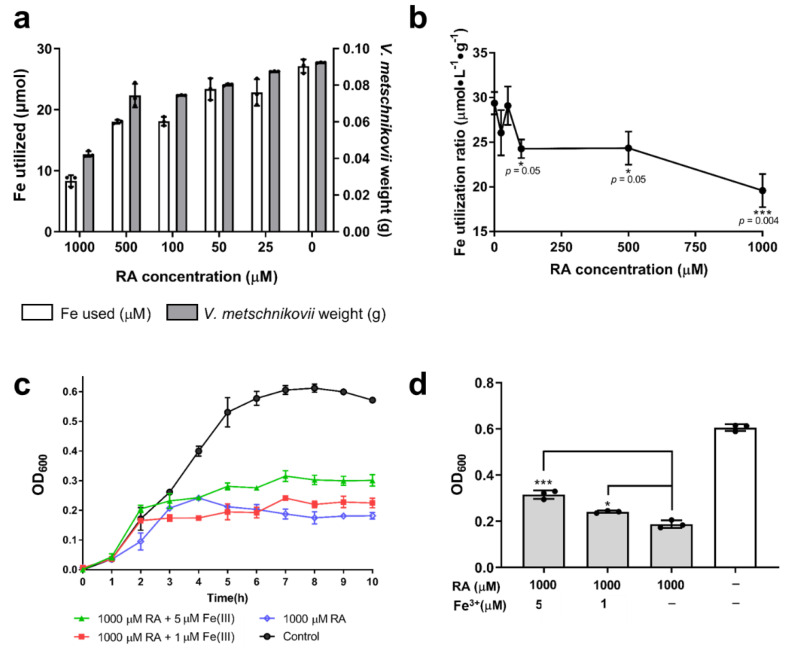
Iron utilization in *V. metschnikovii* and recovery of *V. metschnikovii* by adding Fe^3+^. (**a**) Weight and iron utilization of *V. metschnikovii* after 7 h of culture with RA (0, 25, 50, 100, 500, and 1000 μM). (**b**) Fe utilization ratio of *V. metschnikovii* after 7 h culture with RA (0, 25, 50, 100, 500, 1000 μM). (**c**) *V. metschnikovii* growth from 0 to 10 h in the presence of 1000 μM RA with FeCl_3_ (10 or 50 μM). (**d**) Growth recovery of *V. metschnikovii* at 7 h with 1000 μM RA and supplementation of 1 or 5 μM FeCl_3_. * *p* < 0.05, *** *p* < 0.001.

**Figure 6 ijms-22-13010-f006:**
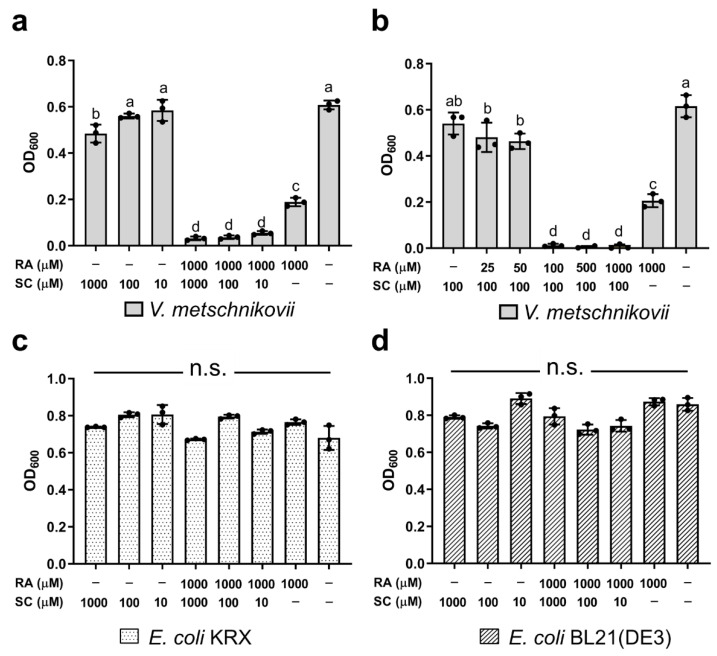
Antibacterial activity of the combination of RA and SC. (**a**) Dose-dependent effects of SC on *V. metschnikovii* inhibition with or without 1000 μM RA. (**b**) Dose-dependent effects of RA with 100 μM SC on *V. metschnikovii* inhibition. (**c**) Dose-dependent effects of SC on *E. coli* (KRX) inhibition with or without 1000 μM RA. (**d**) Dose-dependent effects of SC with or without 1000 μM RA on *E. coli* (BL21(DE3)) inhibition. Means with the same letter are not significantly different from each other (*p* < 0.05). n.s. represents no significant difference.

**Figure 7 ijms-22-13010-f007:**
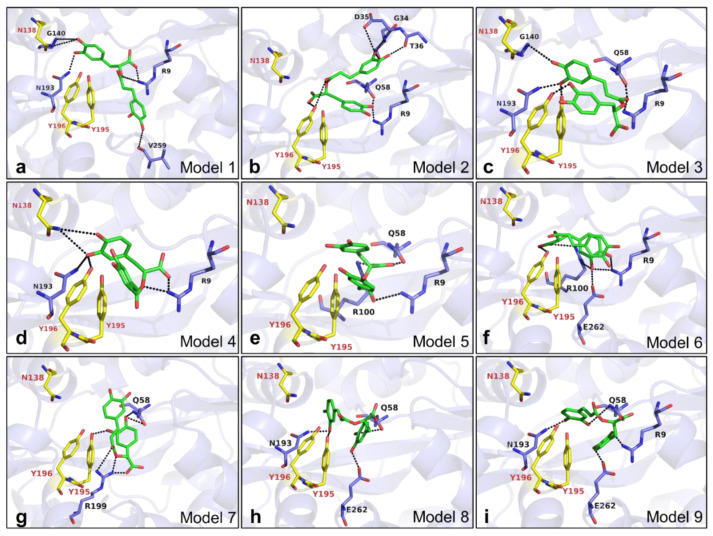
Docking results of RA binding to the Fe^3+^ binding site of VmFbpA. (**a**–**i**) Docking models 1–9 after running docking simulation using AutoDock (version Vina 1.1.2). RA is shown as sticks in green. Fe^3+^ binding site of VmFbpA (N138, Y195, and Y196) shown as sticks in yellow. The amino acid residues predicted to form hydrogen bonds toward RA are shown as sticks in purple. The hydrogen bonds predicted by PDBePISA are shown as black dashed lines. VmFbpA is shown as a cartoon with a transparency of 60% and colored in purple.

**Figure 8 ijms-22-13010-f008:**
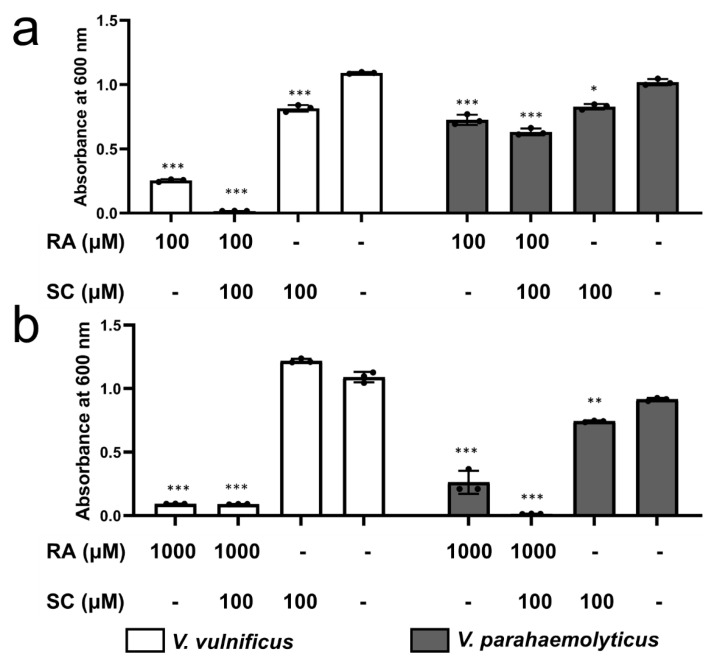
Antibacterial effects of RA and SC on *V. vulnificus* and *V. parahaemolyticus*. (**a**) Effect of 100 mM RA with 100 mM SC compared to RA and SC only. (**b**) Effect of 1000 mM RA with 100 mM SC compared to RA and SC only. The controls did not have exogenous chemicals added. * *p* < 0.05, ** *p* < 0.01, *** *p* < 0.001.

**Table 1 ijms-22-13010-t001:** Comparison of transporters involved in the uptake of different iron sources.

	*E. coli*	*V. metschnikovii*
Fe^3+^	FbpBC/A [25]	FbpBC/A ^a^
Fe^3+^-siderophore	FhuBC/D (Fe^3+^-hydroxamate) [26] FepABC (Fe^3+^-enterobactin) [26] IutA (Fe^3+^-aerobactin) [26] FyuA (Fe^3+^-yersiniabactin) [26]	CeuBC/A (Fe^3+^-enterochelin) ^a^
Fe^2+^	FeoABC [27] EfeOUB [24] MntH [28]	FeoABC ^a^
Fe-citrate	Fec(CD)E/B [29]	N.A.

^a^ The information was obtained from the complete genome sequence (NZ_CP046793.1), which is available at the National Center for Biotechnology Information (NCBI).

## Data Availability

The crystal structure of apo VmFbpA (PDB: 7W3W) has been deposited in the Protein Data Bank (www.rcsb.org).

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
