# Peer review of "Rosmarinic Acid and Sodium Citrate Have a Synergistic Bacteriostatic Effect against Vibrio Species by Inhibiting Iron Uptake"

_ijms, 2021, doi:10.3390/ijms222313010_

Round 1

Reviewer 1 Report

The objectives of the study are clearly outlined. The methods used are described in detail. The results of the experiments are well discussed.

I have just one comment concerning the data visualisation in multi-panel figures. I think it would be more readable if at least some of the figures were shown as separate sets.

For example, the data resulting from binding (ITC) experiments (Fig. 2 g-j). Also, I would use different colour bars for different species i.e. E. coli and V. metschnikovii in Fig.3, and try to arrange the individual facets according to the type of plot to make it straightforward to interpret.

Reviewer 2 Report

The authors present an interesting report of the biologically active compound rosmarinic acid and a synergistic effect in combination with sodium citrate by inhibiting iron uptake in V. metschnikovii specifically. Various chemical and biological techniques are applied to show the bacteriostatic effect and possible molecular mechanism behind it.

Main remarks

The authors selected 20 spices and performed a screening for VmFbpA inhibitors. In figure 1a the remaining Fe concentration is plotted against the spices, including controls. Based on this graph rosemary and cinnamon were selected on their strongest potential of inhibition VmFbpA. The remaining Fe was measured and was around 50% when these spices were used, compared to controls.

In the reviewers opinion the exclusion and inclusion criteria of the selected spices should be better explained, because basil and ginger (and others) also show great potential. Which could be explained by the purity of the extracts and concentration of the specific inhibitor in the extracts. Although basil and ginger do not show ~50% reduction it is still considerably better than garlic and cumin compared to the control.

The inhibition of bacterial growth by RA was evaluated. In figure 3a, a growth curve is shown with different RA concentration ranging from 25 uM up to 1000 uM. This experiment was also conducted on two E. coli strains shown in figure 3c and d. The RA concentration of 500 and 1000 µM show a very large inhibition of growth on the V. metschnikovii but not on the E. coli strains. It is the authors opinion that the inhibition properties of RA on the bacterial growth is the cause, which is further explained in figure 3g where the recovery of growth of V. metschnikovii after adding FeCl3 is shown.

The reviewer would like to know if the inhibition of growth of V. metschnikovii is the only cause of the inhibiting effect of RA and not caused by a decrease in pH of the growth medium (NB ph 7.0), when using the concentration of 500 and 1000 µM RA?

To continue on the previous question, the combinations of RA and SC showed an improvement or synergistic effect. The authors describe the possible mechanism of the iron restrictions in both E. coli and V. metchnikovii and explain this by the different transporter system of Fe2+ and citrate complexes.

The reviewers question is, by adding SC to the growth medium is it possible that the pH would increase and a more alkaline environment is created which stabilizes the RA and therefore is more effective, in contrast of RA alone which might be in a more acidic environment?

In the Introduction it is stated that Vibro infections are increasing and V. metchnikovii was chosen in the experiments because of a potential emerging issues (line 60-62). Although V. cholerae and V. parahaemolyticus are more frequently reported, V. vulnificus and V. parahaemolyticus were chosen to do an alignment of the VmFbpA to other FbpAs and bacteriostatic experiments (line 397-404).

The reviewers question is, why was V. vulnificus chosen instead V. cholerae, which still needed to be confirmed (line 404)?

Comments on the last paragraph of the discussion (line 405-416)

Although the reviewer finds it quite elegant as a finishing touch, to recap history and its food preservative and medicinal use of RA and SC. I do not think it fits in the final paragraph of the discussion and might be better fit for the introduction. Also, V. cholera is mentioned which was not tested in the experiments and the final sentence is a little underwhelming “revealed the molecular mechanism driving the anti-Vibrio activity of umeboshi” in contrast to what the authors actually achieved in their report.

Comment on the conclusion (line 619 to 626)

In the reviewer opinion there are missing a few key conclusions. The authors do not mention anything on possible synergistic possibilities in combination with RA (such as SC). Many references are made in the introduction and discussion towards food preservatives and bacterial growth inhibiting, but none are mentioned in the conclusion, only the reduction of antibiotic use. As a final point, include one or two sentence of what was achieved in the report, such as ‘’molecular mechanism driving the anti-vibro activity’’ (line 415-416).

Minor remarks

Line:

49: A dot is missing after ‘year’.

235: Fig 1. (a) explain the letters above the bars (ef, cd, de, b, bc, g, gf, ef, a).

246: Fig 2.  g, h ,I are too small to read.

249: (e) ‘of’ should be removed here.

256: Fig 3. I, j explain the letters above the bars (b, a, d, c)

277: Table 1. References are missing of the transporters in the table.

303: Fig. 2f should be Fig. 1g.

413: V. colerae should be V. cholerae

570-618: A sterility control is advised for the bacterial stocks, by growing for example on bloodagar you can exclude possible contamination. With the current method it is unknown what micro-organisms might be growing.

Supplementary:

Page 10, figure S4a: Was this the best picture available? The black shade on the bottom does not do justice to the image.

Round 2

Reviewer 2 Report

The authors have sufficiently addressed the comments. To my opinion the article can be published in its current form